# Assessing the Damage Tolerance of Out of Autoclave Manufactured Carbon Fibre Reinforced Polymers Modified with Multi-Walled Carbon Nanotubes

**DOI:** 10.3390/ma12071080

**Published:** 2019-04-02

**Authors:** Polyxeni Dimoka, Spyridon Psarras, Christine Kostagiannakopoulou, Vassilis Kostopoulos

**Affiliations:** 1Department of Mechanical Engineering and Aeronautics, Applied Mechanics Laboratory, University of Patras University Campus, 26504 Patras, Greece; dimoka@mech.upatras.gr (P.D.); s.psarras@mech.upatras.gr (S.P.); kostagia@mech.upatras.gr (C.K.); 2Institute of Chemical Engineering Sciences, Foundation for Research and Technology Hellas (ICE-HT/FORTH), 26504 Patras, Greece

**Keywords:** CFRP, compression after impact, damage tolerance, multi-walled carbon nanotubes, sizing agent

## Abstract

The present study aims to investigate the influence of multi-walled carbon nanotubes (MWCNTs) on the damage tolerance after impact (CAI) of the development of Out of Autoclave (OoA) carbon fibre reinforced polymer (CFRP) laminates. The introduction of MWCNTs into the structure of CFRPs has been succeeded by adding carbon nanotube-enriched sizing agent for the pre-treatment of the fibre preform and using an in-house developed methodology that can be easily scaled up. The modified CFRPs laminates with 1.5 wt.% MWCNTs were subjected to low velocity impact at three impact energy levels (8, 15 and 30 J) and directly compared with the unmodified laminates. In terms of the CFRPs impact performance, compressive strength of nanomodified composites was improved for all energy levels compared to the reference material. The test results obtained from C-scan analysis of nano-modified specimens showed that the delamination area after the impact is mainly reduced, without the degradation of compressive strength and stiffness, indicating a potential improvement of damage tolerance compared to the reference material. SEM analysis of fracture surfaces revealed the additional energy dissipation mechanisms; pulled-out carbon nanotubes which is the main reason for the improved damage tolerance of the multifunctional composites.

## 1. Introduction

Carbon fibre reinforced polymer (CFRPs) composites are increasingly used as advanced materials in many structural applications that is, aerospace, automotive, marine and so forth, due to their superior in-plane mechanical properties, high specific strength and stiffness and corrosion resistance compared to conventional metals. Because of the poor through-the-thickness properties of composites, they are highly susceptible to impact damage due to brittle matrix behaviour and poor impact damage tolerance. In the aviation industry the structural damages of fibre reinforced polymers (FRPs) are inevitable. This type of damage can occur during aircraft maintenance (e.g., tool drop) or during service life (i.e., bird strike, runway debris, hail impact etc.). The damage tolerance as the main design criterion has been established for years in aviation due to the lack of knowledge about the material’s impact behaviour and the existing difficulty of determining the external visibility of impact damages of structures. In high velocity impacts the damage can be easily recognized by the naked eye whereas low velocity impacts can produce matrix cracks within the plies which lead to the presence of extensive delamination, recognized by barely visible impact damage (BVID). In order to enhance the composite resistance due to low velocity impact and the damage tolerance after impact, the impact damage should be reduced. During an impact event, the incident energy is absorbed by a complex combination of energy absorption mechanisms. The presence of matrix cracks within the plies leads to the initial failure during an impact event. These cracks are generated due to through-thickness shear stresses. Delamination as a primary failure mode during low-velocity impact is caused by the extension of matrix cracks. Greenhalgh et al. [1] referred the dominant failure modes that observed in composites during low velocity. More precisely, it was also mentioned the reason where the initiation and the propagation delamination is caused. Particularly, delamination is mainly caused by the interlaminar shear stresses (Mode II) while the presence of fibre breakage, as a significant energy absorbing mechanism, is generated by the high through-thickness forces during impact event. The last few years, several researchers have focused on the improvement of damage resistance and damage tolerance after impact in composite materials [1,2,3,4]. The brittle nature of matrix resins and especially of the epoxy resins as a dominant factor for the susceptibility of composites to delamination was extensively reported to the literature. Several methods have successfully developed for enhancing the impact tolerance of CFRPs such as z-pinning, stitching, interleaving and toughening the matrix using micro-sized particles including rubbery or thermoplastic polymers. Z pinning was found to be a promising method to improve the interlaminar fracture toughness and increase the impact damage resistance of laminates [5]. Zhang et al. [6] reported that the introduction of z-pins at the centre area of samples reduced the damage area by 19–64% depending on impact energy level and specimen’s thickness and reported to greatly increase the compression after impact (CAI) strength close to 45%. An alternative process of increasing the impact damage tolerance of composites is stitching. In Reference [7] Larsson reported that the stitching enhances both the impact resistance and damage tolerance of a prepreg laminate. By employing polyamide interleaves on cross-ply glass/epoxy composite laminates produced by vacuum assisted resin transfer moulding (VARTM), Daelemans et al. [8] reported the improvement of after impact properties where the impact damaged area was significantly reduced close to 50–60% and even at high impact energies.

Recently, attention was paid on the enhancing of fracture resistance of CFRPs by introducing nano-sized particles such as nanofibers, nanoclays and carbon nanotubes. These fillers introduce additional energy dissipation mechanisms and therefore enhance the fracture properties of polymer and composites. Carbon nanotubes (CNTs) are highly potential fillers that have also been utilized by the researchers worldwide due to their extraordinary mechanical, electrical and thermal properties. Additionally, their high aspect ratio and specific surface area in conjunction with high longitudinal modulus make them an ideal candidate incorporating them into resin matrix at low weight concentrations in order to improve the fracture behaviour and therefore the performance of CFRP composite. Kostopoulos et al. [9] introduced 1% CNF in the resin matrix of laminates and observed a significant increase in fracture energy. The extensive bridging effect due to the presence of CNFs was showed in the fracture surfaces which led to the enhanced fracture properties. Kim et al. [10] reviewed that the introduction of 3 wt.% nanoclay reduce the delamination area and substantially improve the residual strength following impact. Ashrafi et al. [11] reported that the impact induced damaged area was decreased by 5% and the compression after impact strength was 3.5% higher of carbon fibre/epoxy resin composites with incorporation of 0.1%wt. functionalized SWCNTs in the epoxy matrix. Kostopoulos et al. [12] has also been investigated the influence of MWCNTs in delamination area experiments were carried out at different impact energies (2, 8, 12, 16 and 20 J). It was found that the incorporation of 0.5 wt.% MWCNTs in the epoxy resin resulted in a 3% reduction of damage area versus baseline material. In another work, Hosur [13] reviewed that the infusion of nanoclay epoxy resin through woven carbon fabric reduced the impact damage.

On the other hand, reduced compression after impact properties in nano-modified composites with carbon nanotubes were reported in the literature because it was assumed that CNTs may act as stress concentrators due to the presence of agglomerates during dispersion, having as a result the possible presence of extensive matrix cracks. Gorbatikh et al. [14] investigated the effect of CNTs on the impact damage resistance incorporating three different epoxy masterbatches. An increase of delamination area was found for all nano-modified samples while the residual compressive strength was either higher or similar to the one observed as reference. This situation indicates that the nano-doped specimens have a higher damage tolerance compared to the baseline material. Pantelakis et al. [15] investigated the synergistic effect of multi-walled carbon nanotubes on the after-impact behaviour of carbon/ epoxy composites revealing a significant increase of the damaged area and a reduction of CAI strength of nano-modified composites. Umer et al. [16] proposed the introduction of graphene oxide by resin infusion process without negligible effect on the impact response of the nano-modified composites. Sadeghian et al. [17] introduced CNFs into polyester matrix to manufacture multiscale glass fibre reinforced composites by resin infusion process at various concentrations (0.5, 1, 1.5 wt.%). The positive effect on the fracture properties was proved with the addition of 1 wt.% CNF, whereas in the case of 1.5%wt. noticeable CNF filtration effect in the thickness direction and the presence of micro-void formation in the specimens were observed due to the re-agglomerations. It was proved that the increase of volume fraction of nano fillers leads to the formation of agglomerates. Additionally, the increase of resin viscosity due to the incorporation of nano fillers is a common situation and leads to an uneven nanofiller distribution into the manufactured component and/or filtering effects that block the resin close to the inlet gates.

To overcome the existence of the resin filtering problem in OoA processes, studies have focused on the developing of the hybridization methods by chemical vapor deposition (CVD), coating or chemically grafting CNTs on the reinforcement surface. More precisely, Wu et al. [18] grafted 3-aminopropylltriethoxysilane (APS) and CNTs onto the CFs concluding that the impact strength was increased by 33%. However, these techniques can cause the reinforcement strength because of the extreme conditions, as reported in References [19,20].

The manufacturing procedure of multiscale carbon fibre reinforcement using an OoA alternative methodology as a promising approach for the scalable and effective integration of MWCNTs in a form of enriched -sizing agent was proposed. A water-based solution with a given wt.% MWCNTs content was prepared and the carbon fibre woven reinforcement was passed through it at a given rate. Then the fabric was used in its wet form to prepare the fibre preform and the final wet preform was placed in to the mould and dried. During the nano-modification treatment of the woven fabric up to the application of infusion process all the steps have been performed in a closed hood equipped with the appropriate nano-filter under-pressure. Reference and nano-modified CFRP laminates with MWCNTs concentration of 1.5 wt.% were produced by liquid resin infusion (LRI) process. In the current study, the compression after impact behaviour of quasi-isotropic laminates, which were modified with carbon nanotubes-enriched sizing agent on the main reinforcement surface, was further investigated. The laminates were carried out to drop tower impact tests at energy levels ranging from 8 to 30 J and non-destructive C-scan technique was performed to evaluate the damage accumulation after impact. Finally, the quantified of damage tolerance of composites was evaluated through the impact and compression after impact test. The same tests were also performed for the reference material.

## 2. Materials and Methods

### 2.1. Materials

In the present study, the matrix material used was a three-part epoxy system provided by Resoltech Advanced Technology Resin (Rousset, France) [21]. This system consists of an epoxy resin (Resolcoat 1400), an anhydride hardener (Resolcoat 1407) and an imidazole accelerator (AC140) which, as recommended by the manufacturer, are typically mixed in ratios of 100:90:0.5 by weight. The nominal curing cycle was 4 h at 80 °C, followed by post cure for 4 h at 140 °C [22]. The primary reinforcement phase was a twill weave carbon fabric, supplied by Fibermax Composites (Volos, Greece) [23], with an areal density of 194 g/m^2^, with 3K filaments in the fibre bundle and a dry ply thickness of approximately 0.35 mm. The Pyrofil TR30S fibre type is a high strength and high modulus aerospace fibre with a reported tensile strength up to 4410 MPa, an elastic modulus close to 235 GPa and density of 1.79g/cm^3^ [24]. Regarding the nano-modified solution, a liquid CNT-enriched sizing agent (SIZICYL XC R2G) was supplied by Nanocyl SA (Sambreville, Belgium) [25]. It is a sizing agent modified with 6.2% maximum solid content of MWCNTs (NC7000, Nanocyl) to size fabrics before impregnation by the resin matrix. The MWCNTs have average diameter and length of 9.5 nm and 1.5 μm, respectively [26].

As the preparation of the nano-modified suspension is concerned, an amount of MWCNT-enriched sizing agent with given wt.% CNTs was diluted into a suitable amount of distilled water obtaining a water solution with 1.5 wt.% MWCNTs. The mixture was subjected to ultrasonic processing with a tip sonication device (Bandelin, Berlin, Germany) to disperse nano fillers in suspension for 3h, operated at an output power of 60W. In order to avoid the temperature’s increasement of suspension during ultrasonic dispersion, the mixing pot was placed into a temperature control bath with continuous fresh water supply. The quality control of the mixture was performed utilizing a grindometer to confirm the absence of agglomerates. Finally, prior to pre-treatment of carbon fabric reinforcement, the prepared water-based solution was degassed in a vacuum chamber for about 5 min.

Quasi isotropic [(+45/−45)/(0/90)]_5s_ carbon fibre reinforced laminates (CFRPs) 350 × 350 mm^2^ were manufactured using twenty plies by Vacuum Assisted Resin Infusion process (VARI). The step by step preparation of doped CFRP laminates is schematically depicted in Figure 1. In the case of nano-doped laminates, the aforementioned water based nano-modified suspension was used for the pre-treatment of all dry carbon fabric layers in a closed hood equipped with the appropriate nano-filters under slight under-pressure. The fabric reinforcement was passed through the solution at a given rate and slightly compressed at the output of the solution bath with a view to produce pre-treated fabrics. Then, the nano-modified wet fabrics were left to dry using a heating plate in a closed hood equipped. To calculate the amount of final CNTs deposited on the carbon fabrics and therefore the received CNT content, the CF plies were weighed before and after deposition. Then, dry nano-modified carbon fabric layers were stacked on a tool, covered by a highly permeable medium and sealed with vacuum bag. The epoxy resin was infused through the preform under vacuum of 660–710 mmHg and the laminates were left to cure in a conventional oven at 80 °C for 4 h followed by post-cure profile at 140 °C for 4 h according to manufacturer guidelines.

The infusion stage was processed at the same temperature conditions for each type of laminates, as depicted in Figure 2. FLIR infrared camera (type T335, FLIR, Taby, Sweden) monitored the heating process during the resin injection. The reference laminates were also produced without the pre-treatment step of carbon fibre preform. The fibre volume fraction and thickness of all produced laminates was around 57% and 4 ± 0.03 mm, respectively, as mentioned in Table 1. In total, nine specimens of 150 × 100 mm^2^ were extracted from the panels for each material type.

From our previous study regarding the influence of carbon nanotubes on the fracture toughness of CFRP composites, composite plates with various CNTs concentrations (0, 0.5, 1, 1.5 and 2.5 wt.%) were manufactured using the aforementioned innovative manufacturing technique, as described previously. In the case of nano-doped laminates, four nano-modified carbon fabric plies for each filler content were placed in the middle of the laminates where PTFE film was placed in the middle plane of each plate to generate the starter crack. Then Mode I and Mode II fracture toughness tests were carried out according to ASTM D5528 [27] and ASTM D7905 [28] standard. The results showed significant increase of about 100% and 60% both in Mode I and Mode II fracture toughness respectively of nano-modified specimens in case of 1.5 wt.% MWCNTs [29]. The fracture toughness of modified composites showed no significant difference with respect to the reference composites in the lower MWCNTs content, under the 1.5 wt.%.

### 2.2. Work methods

#### 2.2.1. Low Velocity Impact Test

The low velocity impacts were performed according to ASTM D7136 standard [30]. This test method determines the damage resistance of composites subjected to a drop-weight impact event without penetration. An in-house built drop weight impactor (AML Group, Patras, Greece), equipped with a 16 mm diameter hemispherical hardened steel striker tip was used for this purpose. The specimens were fixed with the use of four clamps on each side and three cases of low energy impact level 8 J, 15 J and 30 J were carried out. Three specimens with dimension of 150 × 100 × 4 mm^3^ for each material type and for each of three energy levels (8, 15, 30 J) were tested. Different weights and initial heights values of the impactor were suitably selected for each energy level. Specifically, the weights of the impactor were adjusted to 1.22 kg, 1.72 kg and 3.22 kg and the initial heights were pre-determined to 0.67 m, 0.87 m and 0.95 m for each of three energy levels: 8, 15, 30 J, respectively. The measured velocities just before impacts for each of the aforementioned energy impact levels were 3.65, 4.15 and 4.28 m/s, respectively. The impactor was manually left to all from the pre-determined drop height and arrested after rebounding to avoid a sequential strike. After every impact test, the composite specimens were examined for visible damage, detecting possible delamination and extended fibre cracking caused by the impact.

#### 2.2.2. Compression after Impact Test

Subsequently, all impacted specimens were subjected to compression after impact (CAI) tests according to ASTM D 7137 standard [31]. For this purpose, an anti-buckling jig was employed to support the specimen edges and to prevent from out-of-plane displacements. An Instron hydraulic universal testing machine (Instron 8802, Athens, Greece) with a 250 kN load cell was used and a constant crosshead speed was kept at 0.5mm/min, until failure. For the compression test, the specimen was placed in the fixture and a compressive force was applied until a maximum force was reached and then the load suddenly drops. This fixture plays an important role for the compression after impact test results as it minimizes the specimens’ buckling during test, Figure 3a. The load drop was related to the final failure of the specimen in its centre area where the impact event place, as depicted in Figure 3b.

#### 2.2.3. Non Destructive Inspection (C-Scan)

Quality control for all produced laminates was carried out utilizing the C-scan ultrasonic technique. The equipment consists of a MISTRAS Group AD-IPR 1210-PCI card (MISTRAS Group, Athens, GREECE) and a VUB2000 tank (MISTRAS Group). The transducer was a Krautkramer single element probe (MISTRAS Group) at 5 MHz, non-focal. The results showed fully acceptable quality without major defects based on colour bar (thickness variations, inhomogeneities and porosity). The green areas indicate more sound attenuation and therefore lower quality of the part while the red areas indicate less sound attenuation and higher quality of the part without defect or damage being detected. All test samples before and after impact as well as after CAI tests were conducted to NDT (C-scan) in order to evaluate the extent of damage.

#### 2.2.4. Optical Microscopy and SEM

According to the optical microscopy (OM) analysis (ToupCam, Hangzhou, Zhejiang, China), the centre cross sections, where the impact strike has taken place, both for the reference and the nano-modified specimens were examined after impact and compression after impact tests. Additionally, the fracture surfaces of the tested specimens at same centre area only with MWCNTs were examined using a FEI InspectTM F50 scanning electron microscopy (SEM, Patras, Greece). OM and SEM examinations were performed in order to confirm the influence of multi-walled carbon nanotubes that they were responsible for the enhancement of compression after impact properties due to the additional toughening mechanisms that they introduced.

## 3. Results

### 3.1. Low Velocity Impact Test

The dominant failure modes, such as delamination, matrix cracking and fibre breakage, were observed in the composites after the low velocity impact test where these deformations are visible through naked eye. A similar type of damage is exhibited in both of two materials, reference and nano-modified specimens. Figure 4 depicts the damage areas of doped samples for all energy levels from the impacted and the back side of specimens. The visible observation by naked eye of the damage on the specimen surface indicated that all samples of each material type fractured in a similar way. The damaged areas of specimens after impact were measured by an image analysis software (Image J, 1.52a, Bethesda, Rockville, MD, USA) [32] in order to quantify the extent of damage areas in the front and the back side of specimens for each impact energy level, as shown in Figure 4. It is well observed, as the impact energy level was increased, the damage area was increased. Specifically, a visible dent has been formed in the cases of 15 J and 30 J on the front side, which the impactor came in contact with the sample during impact. In the case of impact energy of 15 J there was a visible minor dent depth at the front side of specimen while extensive front and back-face damage was observed at 30 J. On the back side, fibre breakage was obvious only in the cases of 15 J and 30 J impact energies. Samples impacted at 8 J did not show discernible damage in the front and back side, when visually inspected with naked eyes but ultrasonic inspection revealed a small damage at the point of impact. It has to be noted that the damaged area was increased when the impact energy was increased for each material system and it was generally much larger in the back side than in the front one, as expected. These failure modes corresponded very well to the review work of Richardson and Wisheart [2].

After the impact, the induced damage was evaluated both for the reference and nano-modified specimens with 1.5 wt.% MWCNTs by non-destructive testing (C-scan) using the same equipment and the parameters described earlier to assess the extent of damage. To evaluate the C-scan results and to quantify the delamination areas, an image analysis software (Image J) [32] was utilized to post-process the C-scan images. According to the quantification of the damage assessment, each damage size can be correlated to the residual strength of the material. The maximum damage length and width were measured in order to determine the impact damage area. Figure 5 illustrates the delaminated areas of a reference as well as a nano-doped specimen before and after impact test. The white regions in the centre of each specimen represent the damaged area induced by the low velocity impact event. These results were verified using the post-processing image software, where the damaged area of each specimen was measured, as depicted in Figure 6. In the case of nano-modified material, the measured delaminated areas tend to be smaller compared to the reference material both for impact energy levels of 8 J and 15 J. The reduction of damage area can directly be related to the contribution of MWCNTs exhibited more resistance to delamination in the direction of maximum interlaminar shear (45°), where maximum delamination is expected to occur. On the other hand, in the case of the higher impact energy, the extent of delamination detected by C-scan runs of the CNT-enriched composites is larger than one would expect from the visible surface damage but results showed large standard deviations. These observations are in agreement with the results reported by Siegfried et al. [14], where similar damage condition was observed for neat and CNT-modified CFRP composites at the same energy level. According to the author, the reason for the increased damage area is due to the presence of agglomerates which act as stress concentrations and initiation sites for delamination.

### 3.2. Compression after Impact Test

C-scan runs after compression-after-impact tests both for reference and nano-modified specimens were carried out, respectively, in order to assess the contribution of nano-particles on the damage failure of composites after compression-after impact test. As can be seen in Figure 5, C-scan runs after CAI tests for each material type and for all energy levels present that the failure caused by the impacts. As can be illustrated, the failure in composites’ sides close to grips was not observed thus it was concluded that the global buckling of composites was avoided. The failure propagation path of the reference specimens was located at the centre impact area of specimens alongside of specimen’s width. On the other hand, the failure propagation path of nano-modified specimens was observed in different manner. The fracture did not become across the entire width of the nano-modified specimens, as observed in the reference material. The C-scan images indicated that the propagation path of nano-doped specimens was located close to the maximum interlaminar shear direction due to the additional energy dissipation mechanism that introduced by the presence of carbon nanotubes. Thus, in this direction, the lower delamination resistance of the nano-modified specimens was revealed.

The residual compressive after impact strength and the effective modulus were measured for both the reference and nano-modified composites with 1.5 wt.% MWCNTs concentration, according to the referenced standard.

Figure 7a illustrates the typical compressive stress-strain curves for reference and nano-modified CFRP specimens with 1.5 wt.% MWCNTs concentration under compression for each energy impact event (8, 15, 30 J). In the compression after impact test, all specimens in each material type behave similarly and dramatic differences were not observed. The Instron machine was set to stop the test after the sudden drop during compression test which is obtained with a linear drop of the curves after the maximum load was reached. This drop is related to the final failure of the specimen which is triggered by the impact event and is developed across the centre of the specimen in the lateral direction. According to these results, it is concluded that the nano-modified CFRP composites exhibited a small decrease of the compression stiffness at a rate of 3.3% and only 0.8% at impact energy of 8 J and 30 J, respectively compared to reference material and a negligible increase in the case of 15 J, as depicted in Figure 7b.

As it is clearly seen in Figure 7c, an enhancement in the compression after impact strength of the CNT-modified CFRPs with the addition of 1.5 wt.% onto the woven fabric reinforcement was in the range of 8–18% for the different impact energy. It is worth to mention that taking into account the delamination areas (see Figure 6), in the case of the higher impact energy (30 J) the residual strength of the impacted nano-modified specimens did not degrade although the damaged areas in this material type were increased compared to the reference ones. This fact indicates that the nano-modified laminates have a higher damage tolerance than the reference material. It appears that the network-like structure of the CNT dispersion in the woven reinforcement has a beneficial effect on the residual compressive strength. Specifically, the compressive strengths of the reference material were 259 ± 16 MPa, 224 ± 3 MPa and 190 ± 9 MPa for the energy levels of 8, 15 and 30 J respectively. The compressive strengths of the nano-modified material were 305 ± 9 MPa, 251 ± 0.15 MPa and 204 ± 16 MPa for the energy levels of 8, 15 and 30 J respectively. Therefore, the increase in CAI strength of the nano-modified CFRPs was around of 18%, 12% and 8% for the impact energy of 8 J, 15 J and 30 J, respectively. Yokozeki et al. [33] reported a 7% increase in CAI strength for nano-enhanced laminates with the incorporation of 10wt.% of cup-stacked CNT. Kostopoulos et al. [12] reported a 12–15% increase in residual compressive strength for nano-enhanced prepreg laminates containing 0.5 wt.% MWCNTs compared to baseline.

### 3.3. Optical Microscopy and SEM

The damage morphology inspection of the two material systems was carried out using an optical microscopy for all energy levels. The examined area of interest is the centre cross section of specimens where the impact strike has taken place, as Figure 8 depicts. OM analysis was examined both for the cross sections after impact and compression after impact tests, as shown in Figure 8a,b, respectively. Specifically, OM examination for CFRP samples of centre cross section after impact tests present that no damage was detected close to the impacted side of specimens both for two material systems (i.e., fibre fracture, matrix cracks etc.), as depicted in Figure 8a. In general, the impact energy mainly absorbed in the form of elastic and plastic deformation and from other failure modes. More precisely, it is worth to mention that the deformation was not observed for composite materials.

In the case where the optical microscopy assessment was done after the compression after impact test, the main failure modes (delamination, matrix cracks and fibre breakage) are more visible, as shown in Figure 8b. Additionally, each specimen was examined in the area out of the boundary of the impact where matrix cracks could not be found. For that reason, it could be concluded that these matrix cracks are damage due to the impact event and not a result of the compression test. In terms of the reference material, multiple delaminations between plies, matrix cracks and extensive cracks inside tows were found. These matrix cracks were inclined at about 45° and followed the conical shape of the impact damage. In terms of the nano-modified specimens, only extensive fibre breakages and a lesser presence of matrix cracks were observed compared to the reference material. Based on this situation, we can conclude that the matrix of the reference material is weak under a compressive load caused by the impactor. In addition, it is worth to mention that the matrix cracks observed by optical microscopy have no severe effect on the compressive stiffness of the laminates but they also act as an initiation mechanism for the delamination. Additionally, it could be reported that the impact resistance of the nano-modified composites could be improved by the addition of carbon nanotubes on to the carbon fabric surface for all impact energy levels because of the absence of the extensive matrix cracks and delamination. Furthermore, the measured damage areas were smaller than the reference composites in the case of the impact energy of 8 J and 15 J. In the case of the larger impact energy (30 J)—where the damage area was larger compared with the reference composite and the compressive strength did not deteriorate—the toughening mechanisms that have been introduced by carbon nanotubes such as pull-outs could be responsible for the improvement of the compressive strength and stiffness.

All specimens were examined close to the centre area where the impact event took place, after CAI test. SEM analysis was used in order to confirm the additional energy dissipation mechanisms introduced by MWCNTs that were responsible for the improvement of after impact properties and therefore the performance of composites. Figure 9 presents the failure mechanisms that were observed in the cases of three energy levels of the nano-modified specimens with 1.5 wt.% MWCNTs concentration. By analysing the micrographs captured at different magnifications, carbon nanotubes were distributed uniformly into the material’s structure without the presence of MWCNT agglomerates. The white arrows in these figures indicate the pull-out toughening mechanisms that were activated with the introduction of CNTs at all energy levels. In the area with nanotubes, the fracture surfaces appear to be rougher, indicating an additional energy dissipating mechanism. Carbon nanotubes provide a large interface area, absorbing larger amounts of energy than the reference material resulting in a reduced delamination area for the impact energies of 8 and 15 J. In the case of impact energy of 30 J, a possible explanation for the situation of the larger damaged area of nano-doped specimens could be a lesser extent of MWCNTs, as it is clearly seen in Figure 9c. Concluding, the increased damage severity caused by the impact has not resulted in a reduction of CAI strength and stiffness for the nano-modified material, as previously noted.

## 4. Conclusions

In the present work, MWCNT fillers were utilized for the development of nano-modified hierarchical carbon fibre/epoxy laminates with 1.5 wt.% MWCNTs concentration manufactured by in-house developed methodology. Quasi-isotropic CFRP laminates were produced using vacuum assisted infusion process. The produced composites were subjected to low velocity impact tests at three impact energy levels (8, 15, 30 J), non-destructively examined using C-scan and finally tested under CAI test. The residual strength was improved for all energy impact levels with the incorporation of 1.5 wt.% MWCNTs on to the reinforcement surface compared to reference material. This concentration of carbon nanotubes was highly effective for enhancing the damage tolerance after impact. The effective compressive modulus of nano-modified laminates showed no significant difference with respect to the reference ones but did not deteriorated. An increase of delamination area was only found for the nano-modified laminates at the higher energy level, however a large standard deviation was observed. On the other hand, in the case of impact energy level of 8 J and 15 J, the damage areas of nano-doped composites were decreased, observing small deviation compared to the reference material. The fact that the nano-modified composite plates have a higher residual compressive strength, despite the larger delamination area observed in the impact energy of 30 J, indicates that the nano-modified CFRPs exhibit a higher damage tolerance against the reference laminates. The extensive fibre pull-out of the carbon nanotubes during compressive loading is the main mechanism that contributed to the aforementioned improvements due to the additional energy dissipation mechanisms that introduced. This indicates that the CNT network can have a positive influence on the properties of composite.

## Figures and Tables

**Figure 1 materials-12-01080-f001:**
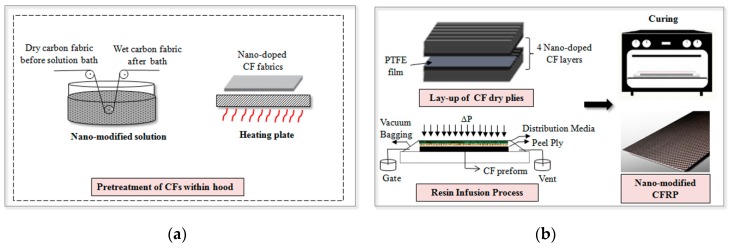
The step by step manufacturing process of nano-modified carbon fibre reinforced polymer (CFRP) laminates. (**a**) The pre-treatment of dry carbon fabrics to produce nano-modified reinforcement, (**b**) The production of CFRP laminates using infusion process (VARTM).

**Figure 2 materials-12-01080-f002:**
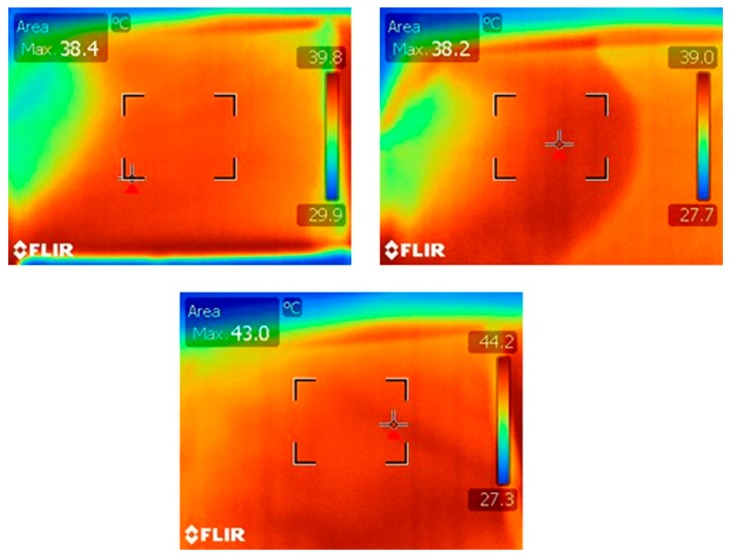
Infrared camera image of resin injection during infusion process.

**Figure 3 materials-12-01080-f003:**
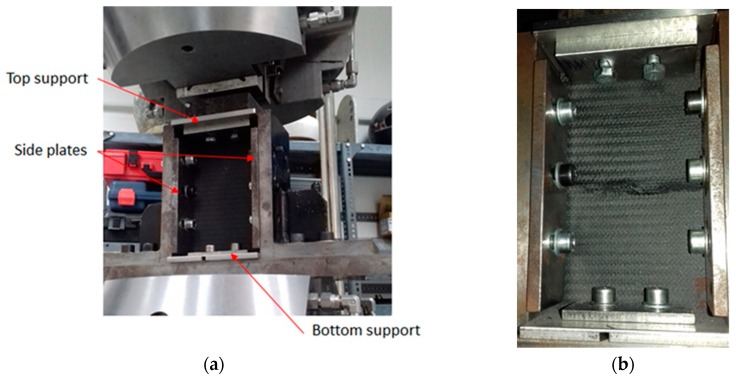
Illustration of (**a**) Compression after impact test fixture with a specimen positioned in place and (**b**) a valid broken specimen after the damage tolerance after impact (CAI) test.

**Figure 4 materials-12-01080-f004:**
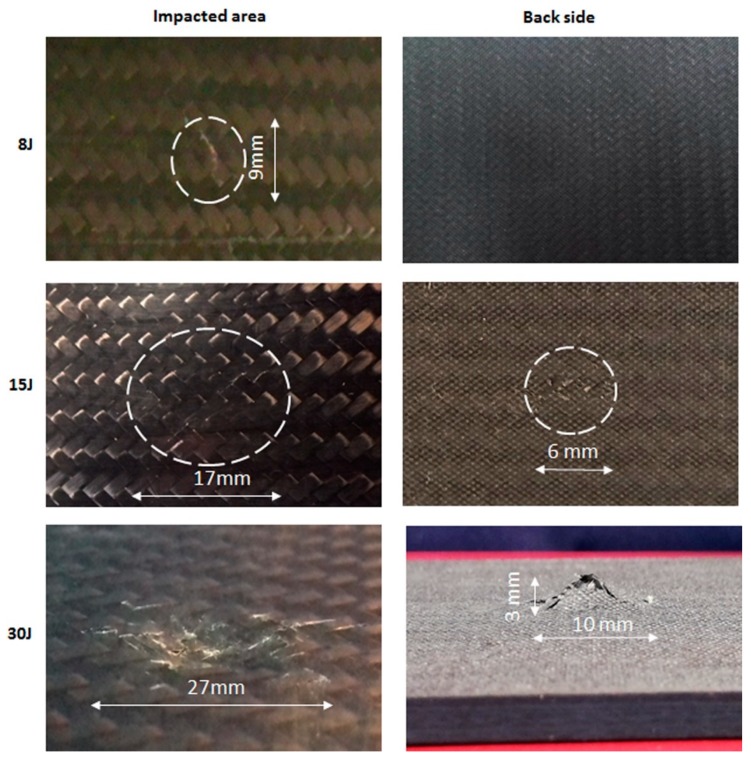
Failure modes after low velocity impact test at impacted and back side of nanomodified specimens with 1.5wt.% multi-walled carbon nanotubes (MWCNTs) at three energy levels.

**Figure 5 materials-12-01080-f005:**
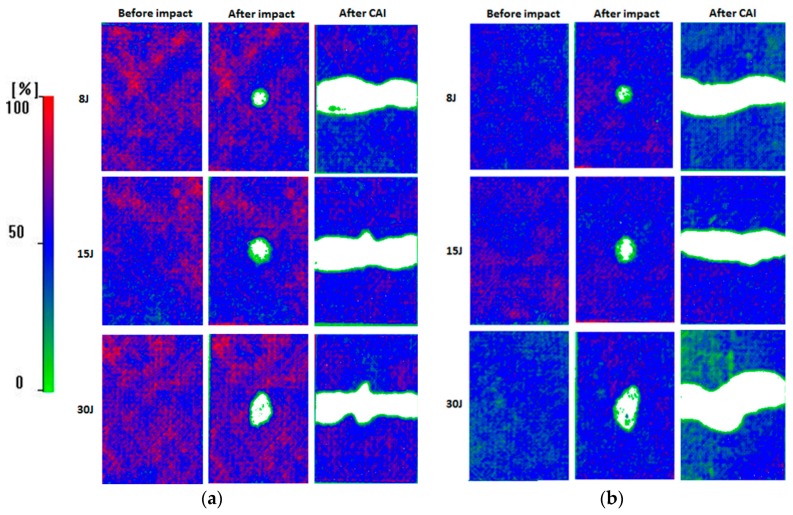
C-scan inspection images before, after impact test and after compression after impact for all energy levels for (**a**) reference material and (**b**) nano-modified with 1.5 wt.% MWCNTs concentration.

**Figure 6 materials-12-01080-f006:**
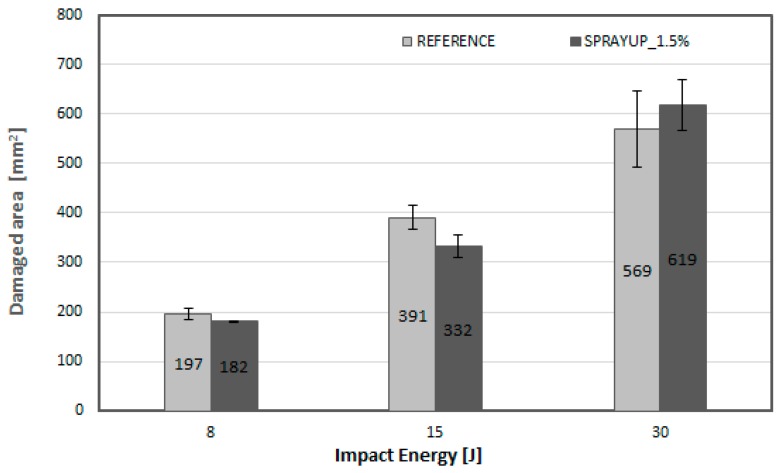
Delamination area versus impact energy levels for both the reference material and the nano-modified with 1.5 wt.% MWCNTs concentration.

**Figure 7 materials-12-01080-f007:**
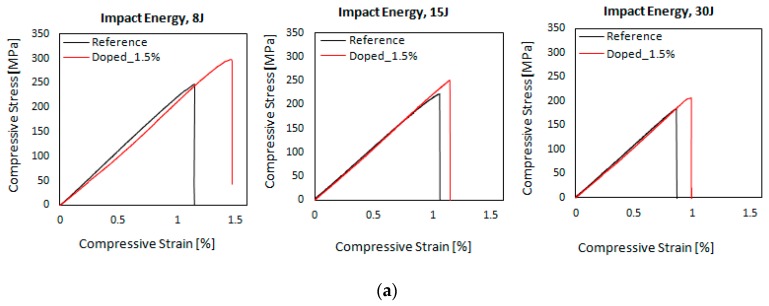
(**a**) Typical compressive stress versus compressive strain (%) curves resulted from CAI test for three energy impact levels (8, 15, 30 J) both for reference and nano-doped CFRPs with 1.5 wt.% MWCNTs concentration, (**b**) Maximum compressive residual strength versus impact energy of the reference and modified CFRPs and (**c**) Compressive modulus vs impact energy of the reference and modified CFRPs.

**Figure 8 materials-12-01080-f008:**
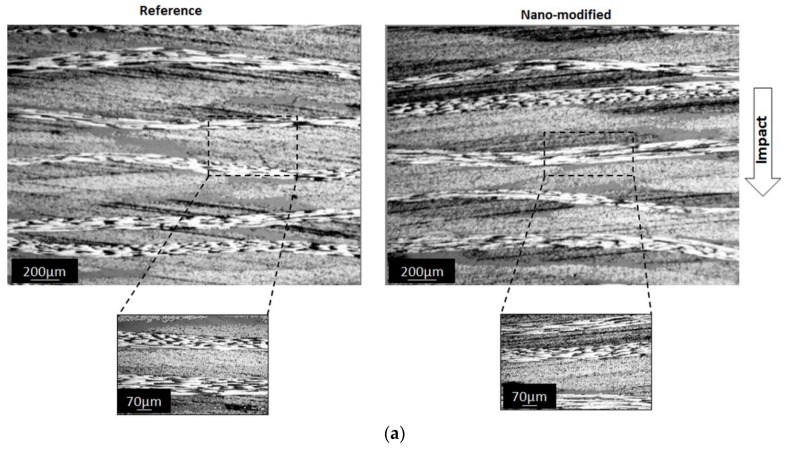
Optical microscope images (**a**) after impact and (**b**) after CAI of reference and nano-modified enriched with 1.5 wt.% MWCNTs specimens at the centre area of impact and at an energy level of 8 J.

**Figure 9 materials-12-01080-f009:**
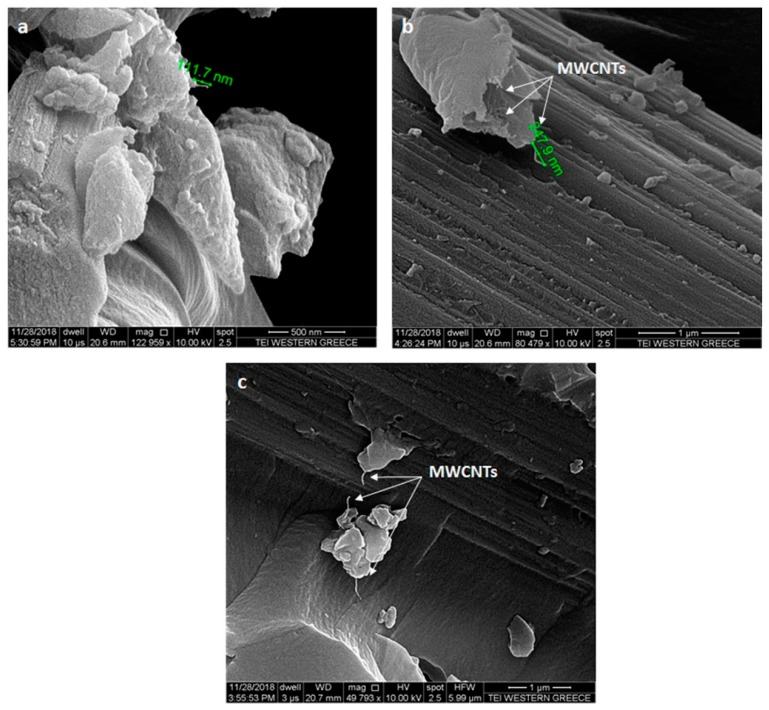
Scanning electron microscope (SEM) micrographs of impacted specimens with 1.5 wt.% MWCNTs at different magnifications at three impact energy levels (**a**) 8 J, (**b**) 15 J and (**c**) 30 J.

**Table 1 materials-12-01080-t001:** Thickness and fibre volume fraction of produced laminates.

Laminate	MWCNTs Content (wt.%)	Thickness (mm)	Fibre Volume Fraction V_f_ (%)
Plate 1-Ref.	0	3.93 ± 0.04	57
Plate 2-Ref.	0	3.94 ± 0.02	57
Plate 3-P15	1.5	4.05 ± 0.03	56
Plate 4-P15	1.5	4.08 ± 0.03	56

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
