# Peer review of "Assessing the Damage Tolerance of Out of Autoclave Manufactured Carbon Fibre Reinforced Polymers Modified with Multi-Walled Carbon Nanotubes"

_materials, 2019, doi:10.3390/ma12071080_

Round 1
Reviewer 1 Report
Dear authors,
Your manuscript work is interesting from experimental point of view.
The novelty consists in the analysis of the effects of the admixture of the multi-walled carbon nanotubes at CFRP on the compression properties after impact.
The manuscript provides important contributions to the research concerning to the mechanical behavior of the CFRP composite materials and this the subject of research matches with the topic of Materials journal. The conclusions are adequately supported by the data presented in order I recommend the publication of this paper after some major improvements.
In order to improve this interesting manuscript, I recommend some minor changes and improvements as it is shown below.
Firstly, I have some recommendations regarding to the organization and division of the manuscript on the sections. According to the template for Materials journal, the manuscript should be divided in the following sections: Introduction, Materials and Methods, Results and Conclusion.
The section “2. Materials and Methods” may be divided in: “2.1. Materials” and “2.2. Work methods”. The sub-section “2.1. Materials” should contain the text from section “2. Manufacturing processes” of your manuscript.
The section “2.2.Work methods” may be divided in the following sub-sections: “Low velocity impact test”, “Compression after impact test”, “Optical Microscopy and SEM”. These sub-sections must contain only the text that refers to the purpose of each test method, a brief description of the test methods and equipment used, loading schemes. The results obtained in these tests (graphs, photos and related text) and some remarks about the results are included in section “3. Results”.
Other improvements and questions:
-Write the value of the pressure of the resin during the infusion process.
-The impact velocity should be computed in each case because the name of the test is “low velocity impact test”. In my opinion, it is much better to analyze the graphics and photos in function of the impact velocity not in function of the impact energy.
-Fig. 6(a):All stress-strain curves should be shown or explain which curve was chosen for each material.
-Fig. 6(c): Title of the vertical axis should be “Compression strength after impact”.
References:
-Write the year of publishing for ASTM D 7136 [25] and ASTM D 7137 [27].
Author Response
Reviewer 1
We appreciate that the reviewer’s comments. The followings are our point-by-point responses.
1. Regarding the first comment of your review about the organization and divisions of the manuscript sections the section were devided as: “2. Materials and Methods” in the sub-sections “2.1. Materials” and “2.2. Work methods” containing the corresponding texts according to your recommendations. Additionally, the sub-section “2.2. Work methods” is divided in “2.2.1 Low velocity impact test”, “2.2.2 Compression after impact test”, “2.2.3 Non Destructive Inspection (C-scan)” and 2.2.4 Optical Microscopy and SEM”,, as recommended. Furthermore, the section “3. Results” is divided in “3.1 Low velocity impact test”, “3.2 Compression after impact test” and “3.3 Optical Microscopy and SEM”.
2. As suggested by the reviewer, we have mentioned the applied pressure resin value during the infusion process in Lines 171-172 of the revised manuscript.
3. Regarding the computed impact velocities just before low velocity impact for each impact energy level, we have reported the measured velocities in the revised manuscript at Lines 212-214, according to your recommendation.
4. Regarding the stress vs strain diagrams, we have added the right excel legend in order to explain which material type corresponds to each curve. We present the experimental curves thats closer to the average values in order clear comparisons to be made. The deviation for each test is presented in the bar chart diagrams.
5. As suggested by the reviewer about the vertical axis of Figure 6(c), we have modified the title’s name to “Compression strength after impact”. In the revised manuscript, this Figure 6 is Figure 7.
6. In accordance of the reviewer’s comment for the year of publishing of ASTM standards, we have written this at the References section (see Lines 504 and 507).
Reviewer 2 Report
- After analysis the manuscript one question is obvious:
What was the real aim of the manuscript? Manufacturing (as you said: lines 115-117) or impact and CAI properties?
If the manufacturing – the manuscript is off-topic
If the properties of modified composites – please look all comments bellow
Please one more time consider the aim of the manuscript.
- The name of the manuscript should be changed. What for you use the “out of autoclave”? this is totally different process. I mean autoclave and OoA they have many common features. Your method is not related to either autoclave or OoA.
- Please add the table with subject of research (types of samples, parameters like thickness etc.)
- How many samples per test? The statistical methods?
- If you prepared modified composite material the microstructure analysis is necessary. Quantitive and qualitive analysis of microstructures
- If you said that the quality control was based on C-scan please present it in detail in appropriate section. The attenuation through the panels would be valuable
- Impact – force time or force-displacement curves? Such curves are important to observe damage initiation etc. You are able to add such measured curves?
- Fig. 3. - The scale line is necessary on the pictures. But the photographs can not be made from the perspective.
- After impact tests the cross section in impact point would be valuable. Without that the results are obvious and not novel. You can leave one sample after impact and cut it. The rests, you can put to CAI tests.
- Line 246: “an anti-buckling jig was employed” what is jig?
- CAI please present the test stand with the sample. The good CAI of composite is not easy to conduct. The boundary conditions are important. Please present the test stand.
- The figure 6a – why the lines are finished without sharp drop at the end of them? You were not observe the sudden degradation of samples? Do you have a photographs of samples during last stage of tests as well as after tests?
- What about the C-scan after CAI? Without that the results of impact and CAI are not correlated.
- The SEM is not enough to take a good conclusions. Please present the picture of global sample, next show the interested area with zoom and next show the place of SEM observation.
- You said:
“Additionally, it could be reported that the impact resistance of the nano-modified composites could be improved by the (line 304) addition of carbon nanotubes on to the carbon fabric surface for all impact energy levels because of (line 305) the absence of the extensive matrix cracks and delaminations. Furthermore, the measured damage (line 306) areas were smaller than the reference composites in the case of impact energy of 8J and 15J. (line 307).” I can not agree with this. If no matrix cracks or delamination what type of energy absorption mechanisms you were observe? “the toughening mechanisms (line 309)”? what is it actually? The toughening can not be named mechanism.
Please prepare a damage analysis in detailed and present it.
- The after impact properties are the consequences of impact damage (impact damage in composite is the major factor which influence the damage growth during CAI). You did not made an appropriate damage analysis after impact as well as after CAI.
- Your final conclusion that “ the CNT network can have a positive influence on the properties of composite (line 357). Can be true only when you will use many different contents of CNT and compare the results. Please write all conclusions one more time without such huge generalizations.
- After read the manuscript one question is obvious:
What was the real aim of the manuscript? Manufacturing (as you said: lines 115-117) or impact and CAI properties?
If the manufacturing – the manuscript is off-topic
If the properties of modified composites – please look all comments bellow
Please one more time consider the aim of the manuscript.
- The name of the manuscript should be changed. What for you use the “out of autoclave”? this is totally different process. I mean autoclave and OoA they have many common features. Your method is not related to either autoclave or OoA.
- Please add the table with subject of research (types of samples, parameters like thickness etc.)
- How many samples per test? The statistical methods?
- If you prepared modified composite material the microstructure analysis is necessary. Quantitive and qualitive analysis of microstructures
- If you said that the quality control was based on C-scan please present it in detail in appropriate section. The attenuation through the panels would be valuable
- Impact – force time or force-displacement curves? Such curves are important to observe damage initiation etc. You are able to add such measured curves?
- Fig. 3. - The scale line is necessary on the pictures. But the photographs can not be made from the perspective.
- After impact tests the cross section in impact point would be valuable. Without that the results are obvious and not novel. You can leave one sample after impact and cut it. The rests, you can put to CAI tests.
- Line 246: “an anti-buckling jig was employed” what is jig?
- CAI please present the test stand with the sample. The good CAI of composite is not easy to conduct. The boundary conditions are important. Please present the test stand.
- The figure 6a – why the lines are finished without sharp drop at the end of them? You were not observe the sudden degradation of samples? Do you have a photographs of samples during last stage of tests as well as after tests?
- What about the C-scan after CAI? Without that the results of impact and CAI are not correlated.
- The SEM is not enough to take a good conclusions. Please present the picture of global sample, next show the interested area with zoom and next show the place of SEM observation.
- You said:
“Additionally, it could be reported that the impact resistance of the nano-modified composites could be improved by the (line 304) addition of carbon nanotubes on to the carbon fabric surface for all impact energy levels because of (line 305) the absence of the extensive matrix cracks and delaminations. Furthermore, the measured damage (line 306) areas were smaller than the reference composites in the case of impact energy of 8J and 15J. (line 307).” I can not agree with this. If no matrix cracks or delamination what type of energy absorption mechanisms you were observe? “the toughening mechanisms (line 309)”? what is it actually? The toughening can not be named mechanism.
Please prepare a damage analysis in detailed and present it.
- The after impact properties are the consequences of impact damage (impact damage in composite is the major factor which influence the damage growth during CAI). You did not made an appropriate damage analysis after impact as well as after CAI.
- Your final conclusion that “ the CNT network can have a positive influence on the properties of composite (line 357). Can be true only when you will use many different contents of CNT and compare the results. Please write all conclusions one more time without such huge generalizations.
Author Response
Reviewer 2
We appreciate that the reviewer’s comments. The followings are our point-by-point responses.
1) Taking into account the reviewer’s comment regarding the dominant scope of this manuscript, the aim of the present study is to achieve enhanced compression after low velocity impact properties with the directly incorporation of carbon nanotubes into the primary reinforcement phase, as the manuscript’s title presents. We disagree with the change of title’s name.
Additionally, regarding the difference of our manufacturing procedure comparing with out of autoclave processes, we propose a novel technique regarding the introduction of the nanoparticles directly into the fiber reinforcement where the application of LRI process is an Out of Autoclave technique because the curing and post-curing of CFRP laminates was executed in a conventional oven (out of autoclave oven), as mentioned in the manuscript of lines 172-173.
2) As suggested by the reviewer to give additional information about samples’ parameters, we add Table 1 mentioning laminates’ thickness and their fiber volume fraction. However, these details of produced laminates were referred in the manuscript (see the Line: 183).
3) Regarding the reviewer’s comment about the number of samples per test, we have mentioned this in the manuscript in Lines 208-209.
4) Regarding the microstructure of nano-modified material with 1.5 wt.% MWCNTs, we have added information in subsection 3.3 regarding the optical and scanning electron microscopy. For the quantitive analysis of the damage, the delaminated areas were measured using post-processing software (Image J), as referred in the section 3.1 where for the qualitive analysis is depicted in Figure 6 with the difference of damaged areas of nano-modified samples compared to reference.
5) Regarding the reviewer’s comment about the attenuation through panels on C-scan runs, the Figure 5 in the revised manuscript is modified where the color bar is added.
6) Regarding the reviewer’s suggested comment to add such measured force vs time or force vs displacement curves, this is not an impact focused paper and there is no added value.
7) Regarding the scale line on the pictures of delaminated area of samples due to low velocity impact, the measured damaged areas were presented on the pictures of Figure 4.
8) Regarding the cross section in impact point just after impact tests, the center cross sections for each material (reference and nano-modified with 1.5 wt.% MWCNTs) are presented in Figure 8(a).
9) About the anti-buckling jig, jig a wide used term and in our case it refers to the anti-buckling device which is described in ASTM standard and it is illustrated in Figure 3(a) at the revised manuscript. Also, Figure 3(b) shows the failure of a specimen after compression after impact. This failure is observed at the center area of specimen.
10) Regarding the reviewer’s comment for the Figure where stress-strain curves presented, this Figure is modified where the compressive load is suddenly drops after reaching its maximum value.
11) Regarding the suggested reviewer’s comment about the C-scan runs after CAI, these C-scan runs both for the reference and nano-modified specimens were presented in Figure 5 (a) and (b), respectively in the revised manuscript.
12) Regarding the reviewer’s comment about zoom out SEM analysis pictures, the necessary for our study was presented and no possibility of zoom out pictures at the given time.
13) Regarding the reviewer’s comment about the failure mechanisms observed on the nano-modified samples after compression after impact tests, matrix cracks and delaminations were observed in both material types, as it was stated. But in the nano-modified specimens did not observed extensive matrix cracks compared with the reference material, as you can see in the optical microscopy images (Figure 8(b) in the revised manuscript).
14) Regarding the reviewer’s comment about the incomplete impact damage analysis, this is not a paper focusing on impact. For the purposes of this study, we have measured the damaged areas of all specimens for each material type for each impact energy level, as depicted in Figure 4 and Figure 6. Furthermore, we have presented the results regarding the compression after impact properties (compression strength and modulus) of nano-modified material compared with the pristine. Additionally, optical and SEM fractography analysis were also presented.
15) Taking into account the reviewer’s comment regarding our final conclusions about the influence of cnts on the after impact properties, we present interlaminar fracture toughness results of the nano-modified material with various CNT contents from our previous work in the subsection 2.1, but this work is unpublished right now (it is accepted from the editorial board of other journal, but is not yet published). Based on these results, we have decided to carry out LVI and CAI tests only for CNT concentration of 1.5 wt.%, because in this weight cnt content showed significant improvement of fracture toughness (Mode I and Mode II).
Round 2
Reviewer 2 Report
Dear Authors
Manuscript looks much better after correction.
However three major remarks should be improve:
Yous said that the impact was not the aim of the manuscript.
In the manuscript (in the one of the most important part - aim (lines 130-132 page 4)
"The laminates were carried out to drop tower impact tests at energy levels ranging from 8 to 30J and non-destructive C-scan technique was performed to evaluate the damage accumulation after impact"
If you wrote that damage evaluation after impact was performed why the impact was not important in your opinion?
If you are trying to present the damage evaluation after impact then this part should be presented much more precisely.
If the impact is the only a tool for preparing the samples for CAI than it should be written clearly because no damage evaluation after impact is not presented.
2. In the corrected manuscript (lines 120-122 page 3) the aim is still a "production of multiscale carbon fiber reinforcement
121 using an OoA alternative methodology".
Please look at the other manuscripts in the world wide literature about the production methods of materials.
Your work is not a production report but it presented the properties (impact?, CAI) of materials produced by using the described method. The aim should be redefined.
3. The CAI stress - strain curves.
Interesting is fact that the stress was dropped to 0 value through vertical line in one moment. Any damage growth before and after max is difficult to observe. In CAI of composites in quite rare situation (keeping in mind the test stand used in CAI).
All can explain macro analysis after CAI tests. Please present the macroview as well as macroscopic analysis of samples after CAI.
If you will decide to correct above elements, it seems to be that the manuscript will be more complete.
Author Response
Reviewer
We appreciate once again the reviewers’ comments that seriously contribute to the improvement of our work.
The followings are our point-by-point responses.
1) Question: You said that the impact is not the aim of the manuscript. If you wrote that damage evaluation after impact was performed why the impact was not important in your opinion?
Answer:
We modified the title of the manuscript in order to better reflect the aim and the content of the work. Thus, the aim of our manuscript is the investigation of the increase of damage tolerance of nano-modified, out of autoclave manufactured, composites. It is well known that the assessment of the damage tolerance of a composite laminate is based upon its Compression after Impact (CAI) behavior, which is directly related to the improvement of the impact resistance. Towards this direction, further to the modification of the title of the revised manuscript (Round 2), we have changed accordingly the content of the work. Changes have been introduced in our manuscript in Lines: 284-285, Lines: 343-344 and in conclusions (see Lines: 425-426 and 431-434).
2) Question: The aim should be redefined.
Answer:
The answer of comment 2 is clearly given above. The damage tolerance of a composite laminate is quantified and compared against different laminates based on CAI experiments and the Hot-Wet compression tests. In the present work we investigate the increase of damage tolerance of nano-modified composite laminates using CAI experiments, when the impact damage has been introduced into the composite laminate using 8, 15 and 30 Joules of impact energy.
3) Question: The CAI-stress-strain curves. Interesting is fact that the stress was dropped to 0 value though vertical line in one moment.
Answer:
Regarding the reviewer’s comment about the stress-strain curves during CAI tests, the UTM of Instron was set to stop the test after the sudden drop of load during compression test, which is demonstrated to the sudden drop of load to zero. This has also introduced in the revised manuscript in Lines: 3219-331. This typical behavior has been also reported in the works of Kostopoulos et. al [12], Ashrafi et. al [11] and Pantelakis et. al [15], as we have mentioned in our manuscript.